# Effect of PCE on Properties of MMA-Based Repair Material for Concrete

**DOI:** 10.3390/ma14040859

**Published:** 2021-02-11

**Authors:** Jian Han, Lingling Xu, Tao Feng, Xin Shi, Pan Zhang

**Affiliations:** College of Materials Science and Engineering, Nanjing Tech University, Nanjing 211816, China; xulingling1964@163.com (L.X.); fengtao9609@163.com (T.F.); 15195903152@163.com (X.S.); zhangpan19970525@163.com (P.Z.)

**Keywords:** repair material, methyl methacrylate, perchloroethylene, mechanical properties, durability

## Abstract

Methyl methacrylate (MMA)-based repair material for concrete has the characteristics of low viscosity, excellent mechanical properties, and good durability. However, its application is limited due to its large shrinkage. Existing studies have shown that adding perchloroethylene can reduce the shrinkage. On this basis, other properties of modified MMA-based repair materials were tested and analyzed in the present study. The results revealed that the addition of perchloroethylene (PCE) can hinder the polymerization reaction of the system. When CaCO_3_ with a mass fraction of 30% was added, the viscosity of the material was within the range of 450–500 mPa·s, and the shrinkage decreased to approximately 10%. The bending strength of MMA, and MMA modified by PCE, repair materials at 28 days could reach up to 28.38 MPa and 29.15 MPa, respectively. After the addition of HS-770 light stabilizer with a mass fraction of 0.4%, the retention ratios of the bending strength of materials with ratios of P0 and P3 could reach 91.11% and 89.94%, respectively, after 1440 h of ultraviolet radiation. The retention ratio of the bending strength of the material could reach more than 95% after immersion in different ionic solutions for 90 days.

## 1. Introduction

Concrete cracking is a common phenomenon due to its intrinsic nature and the erosion induced by the surrounding environment [1,2], and which should be repaired with repair materials with excellent performance. Repair materials can be roughly classified into three categories, according to composition [3,4]. Inorganic repair materials, include sulpho-aluminate cement [5,6], high alumina cement [7], expansive cement, and magnesium phosphate cement [8,9]. For organic repair materials, the most common are epoxy resin [10,11], acrylate, polyurethane [12,13], and urea-formaldehyde resin [14]. For organic and inorganic composite repair materials, the polymers used in cement mortar are mainly epoxy resin [15], acrylate copolymer emulsion [16], and carboxylated styrene-butadiene emulsion [17].

It was found that when the width of a crack is less than 0.2 mm, the repair effect of using a methyl methacrylate (MMA)-based repair material is ideal. The viscosity of MMA can reach as low as 0.8 mPa·s [18]. Furthermore, MMA has good heat resistance, wear resistance, chemical corrosion resistance, and impermeability [19,20,21], and has a strong bond with cement concrete [22]. However, the volume shrinkage of MMA during polymerization is approximately 21%. This large shrinkage affects the bonding effect. Therefore, it is necessary to improve this shrinkage [23,24].

There are two main ways to reduce shrinkage. One approach is to change the structure of the polymer itself, while the other approach is to add inorganic fillers or low profile additives [25,26]. Li, W.-G. [27] studied the effect of adding alumina and aluminum hydroxide to a MMA-based repair material on the shrinkage ratio, and found that the volume shrinkage of the sample decreased. Ou, Y.-G. [28] studied a system of polyvinyl acetate and styrene to improve the shrinkage of MMA. The results revealed that the shrinkage decreased, but the bonding strength also decreased. Su, Z.-Y. [29] preliminarily studied the modification of MMA material by adding epoxy resin, and found that the volume shrinkage significantly decreased. Mun, K.-J. and Choi, N.-W. [30] used different contents of unsaturated polyester resins as adhesives to study the effect on the properties of expanded polystyrene-based polymethyl methacrylate mortar. The experimental results revealed that a low shrinkage can be achieved. Li, X. [31] mixed MMA with sodium acetate to improve the shrinkage. The results revealed that the shrinkage ratio decreased. However, the viscosity of the repair material increased, and the fluidity became worse at the same time. Yang, Z.-W. [32] added perchloroethylene to a MMA system, and the shrinkage ratio decreased to a certain extent. In addition, experiments on bond strength were carried out. It was found that the adhesiveness was good enough to meet the requirements for repair engineering, but other properties were not studied. In addition to the consideration of shrinkage, other properties of modified MMA-based repair materials should also meet the requirements of repair engineering [33]. Therefore, other properties of modified MMA-based repair materials, such as viscosity, bending strength, ultraviolet aging resistance, and chemical erosion resistance were tested and analyzed in the present study.

## 2. Materials and Methods

### 2.1. Raw Materials

The materials used for the present study were methyl methacrylate (C.P., Shanghai Lingfeng Chemical Reagent Co., Ltd., Shanghai, China), benzoyl peroxide (BPO) (C.P., Shanghai Lingfeng Chemical Reagent Co., Ltd., Shanghai, China) as an initiator, dibutyl phthalate (DBP) (C.P., Shanghai Lingfeng Chemical Reagent Co., Ltd., Shanghai, China) as a plasticizer, N, N-dimethylaniline (DMA) (C.P., Shanghai Lingfeng Chemical Reagent Co., Ltd., Shanghai, China) as an accelerator, and perchloroethylene (PCE) (C.P., Shanghai Lingfeng Chemical Reagent Co., Ltd., Shanghai, China).

The inorganic filler was mainly 1500 mesh heavy calcium carbonate (CaCO_3_). The relevant information about the calcium carbonate is shown in Table 1.

Table 2 shows the ratio of raw materials used in the preparation of the MMA-based repair material.

### 2.2. Preparation of the MMA-Based Repair Material

In the storage process of MMA, a small amount of the inhibitor was added to the MMA to prevent the slow self-polymerization reaction from occurring. In general, the content is less than 0.001% by weight, but this would affect the polymerization reaction. As a result, the polymerization inhibitor should be initially removed. For the method, the raw material of MMA was distilled for 10 min at 50 °C in a water bath. Then, after removing the polymerization inhibitor, the MMA was added to a three-mouth flask together with the PCE, initiator, plasticizer, and other additives, and the polymerization reaction was carried out under stirring at 80 °C in a water bath [34]. The sample preparation device is shown in Figure 1. During the reaction process, it was necessary to determine the viscosity of the system at all times, in order to prevent explosive polymerization. After the reaction had preceded for 35–50 min, the viscosity was relatively high, and the heating was stopped in time. Then, the three-mouth flask was taken out and cooled in cold water, in order to obtain the prepolymer of the repair material.

### 2.3. Testing Procedures

#### 2.3.1. Viscosity

The viscosity of the repair material was measured using a NDJ-1 rotary viscometer (Shenzhen, China). The viscometer equipment is shown in Figure 2. First, the sample was placed in the container. Then, the appropriate rotor and rotational speed were selected. Finally, the motor was started, and the data were read after the pointer was stable. After two parallel tests were measured, the average value was taken as the viscosity. When the sample with the ratio of P0 in Table 2 was prepared, the viscosity was measured at different time points, in order to determine the effect of the reaction time on the viscosity of the repair material. Samples with ratios of P0, P1, P2, P3, P4, and P5 were prepared by adding different proportions of PCE. Then, the viscosity of these different systems was measured at the same time, and compared with each other.

#### 2.3.2. Shrinkage

A small test-tube was measured to obtain the mass m_01_ and volume V_0_. Then, the prepared sample was poured into the test-tube, and the total mass was measured to obtain the mass m_02_. As shown in Figure 3, the test-tube was placed in an oven at 60 °C for 4 h. Then, it was taken out, and placed indoors. After five days, the test-tube was broken, and the sample was taken out. The mass m_1_ was obtained by weighing, the volume V_1_ was measured using the drainage method, and the shrinkage ratio S was calculated according to Formula (1). Two samples of each ratio were taken for the measurement, and their average value was taken as the result.
(1)$$S=\frac{\frac{m_{1}}{V_{1}}-\frac{m_{02}-m_{01}}{V_{0}}}{\frac{m_{1}}{V_{1}}}\times 100\%\;$$

Here:*S*: the shrinkage of the sample, %;*m*_1_: the quality of the sample after curing, g;*V*_1_: the volume of the sample after curing, mL;*m*_01_: the mass of the small tube, g;*m*_02_: the mass of the small test tube + the sample, g;*V*_2_: the volume of the small test tube, mL.

#### 2.3.3. Bending Strength

As a material for repairing cracks, the most important mechanical property was the bending strength [35]. After the accelerator with a mass fraction of 0.5% was added, the prepolymer was evenly stirred and poured into a mold, with a size of 100 × 15 × 5 mm^3^, and cured at 60 °C. Then, the strength test was carried out on a CSS-44020 universal testing machine, (Changchun, China) and the maximum load of bending failure of the material was recorded. The bending strength of the repaired material was calculated according to Formula (2). The samples of each ratio were measured three times, and the average value was calculated.
(2)$$f_{m}=\frac{3F_{max}L}{2bh^{2}}$$

Here:*f_m_*: the ultimate bending strength of the material, MPa;*F_max_*: the maximum load of bending failure of the material, N;L: the distance of force between two points of the material, m;*b*: the width of the section of the material specimen, m;*h*: the height of the section of the material specimen, m.

#### 2.3.4. Durability

The durability of the repair material included the ultraviolet aging resistance, chemical erosion resistance, thermal shock aging resistance, and frost resistance. The first study was the ultraviolet aging resistance. Different proportions of bis (2,2,6,6-tetramethyl-4-piperidyl) sebacate (HS-770) type light stabilizer were added into the samples of P0 and P3. Then, the samples were uniformly stirred, poured into a 100 × 15 × 5 mm^3^ mold, and cured at 60 °C. Afterwards these were placed in an ultraviolet box for 1440 h, taken out to test using the CSS-44020 universal testing machine, and the bending strength was calculated according to Formula (2). Next, these were compared with the blank specimens to obtain the retention ratios of the bending strength. Each group of samples was measured three times and the average value was calculated. The position of the ultraviolet lamp and sample specimen in the ultraviolet box is shown in Figure 4.

Good chemical corrosion resistance can make the repair material firmly bond with old concrete in a humid environment, and this can also prevent water and various ions from causing further erosion of the repair materials [36]. In the experiment, a standard cement–mortar ratio with a water–cement ratio of 0.5 and a mortar ratio of 1:3 was used in a mold 20 × 20 × 20 mm^3^. Then, the P0 and P3 repair materials were applied to six surfaces of the mortar, with a thickness of approximately 1 mm. Afterwards, the samples were placed into water with the NaCl solution with a mass fraction of 5%, and the MgSO_4_ solution with a mass fraction of 5%. Next, these were taken out and weighed after a certain period of time to calculate the change in quality before and after the immersion. The chemical corrosion resistance of the MMA-based repair material was evaluated by comparing this with the blank samples. Each group of samples was measured twice, and the average value was calculated.

The samples of P0 and P3 were prepared and poured into a mold of 100× 15 × 5 mm^3^. Then, a thermal shock cycle experiment was carried out. The thermal shock cycle experiment designed for the present study was carried out according to the following steps. After heating at 105 °C for 30 min, the repaired materials were immediately taken out and placed in a −20 °C freezer for 30 min as a thermal shock cycle [37]. Then, the specimens were tested using the CSS-44020 universal testing machine, and the bending strength was calculated according to Formula (2-2). Afterwards these were compared with the blank specimens to obtain the retention ratios of the bending strength after 300 cycles. Each group of samples was measured three times, and the average value was calculated.

Concrete matrix and repair materials are often damaged due to freezing and thawing. Hence, the ability of the anti-freeze–thaw cycle should be taken into account in the preparation of the repair material [38]. The freeze–thaw cycle experiment designed for the present study was carried out according to the following conditions. The freezing–thawing temperature ranged from −20 °C to 20 °C, and the freezing–thawing time was 4 h each time. The specimens were tested using the CSS-44020 universal testing machine, and the bending strength was calculated according to Formula (2-2), with cycles of 50 times and 200 times, in order to determine the frost resistance. Each group of samples was measured three times, and the average value was calculated.

## 3. Results and Discussion

### 3.1. Viscosity

From the relationship between viscosity and time in Figure 5, the reaction can be roughly divided into three stages: The first stage is before 20 min, the second stage is between 20 min and 35 min, and the third stage is after 35 min. The main processes in the first stage of the system were the initiator being decomposed to free radicals, and these free radicals initiating the MMA monomer. Then, the main chain of polymerization slowly grew. Since the molecular weight was relatively small, the viscosity slowly changed. In the second stage, with the increase in free radicals and chain activation centers, the main chain significantly increased, the molecular weight increased, and the viscosity of the system began to increase. In the third stage, the reaction was violent and gave off a lot of heat, which further promoted the reaction, thereby inducing the viscosity to sharply increase with time. It is noteworthy that the reaction time must be controlled well, in order to prevent the phenomenon of explosion.

In addition, it can be observed by comparing the different curves in Figure 5 that the conversion time of P1, P3, and P5 after the addition of PCE was delayed in the three stages. The time point for the transition from the first stage to the second stage was 25 min for P1, 29 min for P3, and 33 min for P5. The time point when the second stage transitioned to the third stage was 40 min for P1, 42 min for P3, and 43 min for P5. It can be observed that the addition of PCE hindered the polymerization reaction of the system, to some extent. Since there are four chlorine substituted atoms in the PCE molecule, the chlorine atom is much larger than the hydrogen atom. Therefore, a relatively obvious steric hindrance effect would occur when PCE is linked to the main chain of MMA. This reduces the curing reaction rate and reaction exothermic, and makes the curing process more stable, thereby causing the reaction process to become slower [39,40].

### 3.2. Shrinkage

Yang, Z.-W. [32] prepared a repair material by adding PCE to a MMA system, and tested its shrinkage ratio. The experimental results revealed that the shrinkage ratio of the MMA-based repairing material after the addition of PCE decreased, to a certain extent. When the mass of PCE was 10% of the mass of MMA, the shrinkage ratio, which was only 15.25%, was the lowest among all the test results. However, the shrinkage was still relatively high for the repair material, and still needed to be reduced. In addition to changing the structure of the repair material, an often used method to reduce shrinkage is the addition of inorganic fillers. The present experiment investigated the influence of CaCO_3_ on the shrinkage of MMA repair materials, and the results are shown in Figure 6. It can be observed from the figure that the shrinkage of the samples with ratios of P0 and P3 had an obvious decreasing trend with the increase in the proportion of CaCO_3_. The shrinkage ratio of the repair material can be reduced to approximately 10% when CaCO_3_ with a mass fraction of 30% of the mass of MMA is added. Li, X. [31] wanted to add sodium acetate to MMA for shrinkage modification. When the mass fraction of sodium acetate was 3% of the MMA mass, the shrinkage rate was 18%. Compared with that, the effect of shrinkage reduction in the present experimental study was more obvious. However, the addition of CaCO_3_ would definitely change the viscosity of the system [41]. The results for the viscosity of the system after the addition of CaCO_3_ are shown in Table 3. It can be observed from Table 3 that the viscosity of the system increased with the increase in CaCO_3_ ratio. The viscosity of the repair material was within the range of 450–500 mPa·s when CaCO_3_, with a mass fraction of 30%, was added. The viscosity sharply increased when the mass fraction of CaCO_3_ was more than 20%. Hence, the excessive viscosity did not meet the requirements of construction. This requires the consideration of both the shrinkage and viscosity of the repair material.

### 3.3. Bending Strength

Samples with ratios of P0, P1, P2, P3, P4, and P5 were tested for bending strength. The experimental results are shown in Figure 7.

It can be observed from Figure 7 that the prepared repair material had a higher bending strength, and that the strength in the early stage was higher. At three days, the bending strength reached up to 19~20 MPa. However, the bending strength slowly developed in the later stage, and its strength reached up to 27~30 MPa at 28 days. The flexural strength of the PMMA mortar prepared by Mun K J and Choi N W was within 25–35 MPa [30], which is similar to the flexural strength of the repair material studied in the present experiment. The addition of PCE had no adverse effect on the bending strength of the repair material, and the strength was enhanced to a certain extent. However, the increased effect was not obvious. It should be noted that the size of the specimen formed in the present study was 100 × 15 × 5 mm^3^. If the specimen size changes it is necessary to consider the size effect. It is possible for the bending strength of the specimen to decrease with the increase in section size of the specimen.

CaCO_3_ treated with a coupling agent was added as a filler to the MMA-based repair material. This can significantly improve the shrinkage of the material. However, this would have an impact on the mechanical properties of the repair material. CaCO_3_ with a mass fraction of 10%, 20%, 30%, 40%, and 50% of the mass of MMA was added into the samples, with the ratios of P0 and P3, respectively. Then, the bending strength of the samples was tested. The bending strength at 28 days was selected, and the relationship between the bending strength and proportion of CaCO_3_ was determined, as shown in Figure 8.

It can be observed from the Figure that the addition of CaCO_3_ had an effect on the bending strength of the repair material. As the proportion of CaCO_3_ increased, the bending strength of the repair materials obviously decreased. The largest decrease was approximately 25%, when CaCO_3_ with a mass fraction of 50% was added. This was mainly due to the addition of CaCO_3_ to the samples. Although this increased the viscosity of the samples, a relatively large viscosity is not conducive for the dispersion of CaCO_3_ in the system. Hence, the CaCO_3_ must be agglomerated in the system. As the proportion of CaCO_3_ increased, the dispersion effect became worse, and there was more agglomeration in the system. As a result, the bending properties of the resin further decreased [42].

### 3.4. Ultraviolet Aging Resistance

Figure 9 presents the bending strength of the samples when they were irradiated in the ultraviolet box for 1440 h after the addition of the light stabilizer with different mass fractions of the mass of MMA. Figure 10 presents the retention ratios for the bending strength. It can be observed from the Figure that the retention ratios for the bending strength significantly increased after the addition of the light stabilizer. When the mass fraction of the light stabilizer was 0.4%, the retention ratio of the bending strength was above 90%. The reason for this was that the light stabilizer HS-770 is a kind of hindered amine light stabilizer, which can be transformed into nitroxyl radicals, partially under the photooxidation condition. These nitroxyl radicals can capture the active radicals generated in the polymer, thereby inhibiting the occurrence of a photooxidation reaction, and playing a stabilizing role [43]. The strength retention rate of the epoxy mortar repair materials studied by Liu F can reach up to approximately 86% under the condition of ultraviolet irradiation for 1000 h [44]. These two repair materials are similar in terms of anti-ultraviolet aging performance, and the performance was excellent.

In addition, the addition of PCE had a certain degree of influence on the ultraviolet aging resistance of the repair material. Samples with the ratios of P0 and P3 were prepared, and the HS-770 light stabilizer with a mass fraction of 0.4% was added. The four groups of samples were labeled P0, P0 + HS-770, P3, and P3 + HS-770, and placed into an ultraviolet box. The bending strength was tested at different times points. The experimental results are shown in Figure 11. It can be observed from the curve in Figure 11 that the bending strength of the repair material significantly decreased with time when the light stabilizer HS-770 was not added. Among these, the ratio of P3 was more significantly decreased. The main reason for this is that PCE contains four chlorine atoms, which results in a relatively large steric hindrance of the system and poor stability of the generated polymer, when PCE is linked to the main chain of PMMA.

### 3.5. Chemical Corrosion Resistance

Table 4 shows the mass changes of mortar blocks coated with MMA-based repair material, and not coated with MMA-based repair material, before and after soaking in different solutions. It can be observed from Table 4 that the impermeability of the ordinary mortar not coated with the MMA-based repair material was very poor. Furthermore, the water content reached as high as 12.79% after being soaked in clean water for 90 days. The impermeability improved greatly after the application of the MMA-based repair material, and the water content was only approximately 3% after 90 days. Therefore, the MMA-based repair material can form a very thin resin film after curing on the surface of the mortar blocks, and this resin film can effectively prevent the flow of water and various ions, thereby achieving a sealing effect.

The samples with two ratios of P0 and P3 were poured into a mold with a size of 100 × 15 × 5 mm^3^. After five days, the samples were immersed in clean water; the NaCl solution with a mass fraction of 5% and the MgSO_4_ solution with a mass fraction of 5%, respectively. The bending strength after 90 days was tested and compared with the blank samples, and the retention ratios of bending strength were calculated. Each group of samples was measured three times, and the average value was calculated. The results are shown in Table 5. In the table, it can be observed that samples with the ratios of P0 and P3 had a strong resistance to chemical erosion after soaking for 90 days. The retention ratios of the bending strength could reach over 95% in the NaCl solution and the MgSO_4_ solution. The reason for this is that the MMA-based repair material had a compact structure after curing, which could effectively prevent the penetration of liquid and ions. In addition, the MMA repair material is an organic material, which has little interaction with chloride ion, sulfate ion, and other ions. ce, it had a high retention ratio for bending strength in the environment of the salt solution. The water glass suspension double-liquid grouting material studied by Wang H X was soaked in a Na_2_SO_4_ solution for 180 days. The mass loss rate was 5%, while the strength loss rate was approximately 25% [45]. It can be observed from the comparison of the data that the MMA repair material studied in the present experiment had good chemical erosion resistance.

### 3.6. Thermal Shock Aging Resistance

Table 6 shows the retention ratios for the bending strength of the MMA-based repair material after 300 thermal shock cycles. It can be observed from the data in Table 6 that the bending strength of the two ratios, P0 and P3, of the repair materials significantly decreased. They decreased from 28.38 MPa and 29.15 MPa, to 18.55 MPa and 17.85 MPa, respectively, and the retention ratios for the bending strength were only 65.36% and 61.23%, respectively. The thermal shock resistance of the repair material with PCE was worse. Han, Y.-F. [14] conducted a thermal shock resistance cycle test on a modified epoxy resin, and the strength retention rate after 28 days of testing was 67.52%. By comparing these two materials, it was found that the thermal shock aging resistance of both materials was poor. Under the condition of high temperature, the thermal movement of the molecular chain of the MMA-based repair material was violent. Part of the molecular chain was too late in rearrangement when the temperature suddenly dropped, which resulted in fracture, and the mechanical properties of the repair material declined due to repeated cycles. The stability of the polymer with the ratio of P3 decreased due to the presence of PCE in the system, and part of the PCE broke from the main chain at high temperature, making the thermal shock cycle performance worse.

In the present experiment, polypropylene fiber (PPF) with different mass fractions was selected to be added into the samples, in order to improve the thermal shock performance. The bending strength after 300 thermal shocks is shown in Figure 12. The retention ratios for the bending strength were obtained by comparing this with the blank sample. The results are shown in Figure 13.

It can be seen from the figure that the retention ratios of the bending strength of the repaired materials after the thermal shock cycle were significantly improved after the addition of PPF. The retention ratios of the ratios of P0 and P3 increased from 65.36% and 61.23%, to 87.01% and 86.74%, respectively, after adding PPF with a mass fraction of 1.5%. The bending strength reached 24.69 MPa and 25.28 MPa, which were higher than the bending strength of ordinary concrete. It is generally believed that the fiber is surrounded completely by the matrix after being added into the material, and that the stress is transmitted to the fiber through the interface. The strength of the reinforcing fiber is much higher than the strength of the covering layer, thereby improving the crack resistance and strength of the material [46].

### 3.7. Thermal Shock Aging Resistance

Table 7 shows the experimental results after the freeze–thaw cycle test. It can be observed from Table 7 that the repair materials with the ratios of P0 and P3 had a good freeze–thaw cycle resistance. For these, the retention ratios for the bending strength of the repaired materials after 50 freeze–thaw cycles reached more than 98%, and the retention ratios for the bending strength after 200 freeze–thaw cycles also reached more than 95%. The reason for this is because MMA is relatively dense after complete curing. This prevents water from entering to a certain extent, and prevents the damage from frost under low temperature conditions. In addition, the properties of the polymer repair material are relatively stable at low temperatures. Han, Y.-F. [14] tested the frost resistance of modified epoxy resin, and the strength retention rate was 60.68% after testing for 15 days. Compared with that, the frost resistance of the MMA repair material was excellent.

## 4. Conclusions

The addition of PCE can hinder the polymerization reaction of the system to a certain extent, and make the reaction rate slow down.The addition of CaCO_3_ can effectively reduce the shrinkage of the MMA-based repair material, and this can also increase the viscosity of the repair material. The viscosity of the material is approximately 500 mPa·s, and the shrinkage ratio can be reduced to approximately 10% when CaCO_3_ with a mass fraction of 30% is added.The MMA and MMA modified by PCE repair materials both have good mechanical properties, and the bending strength at 28 days can reach up to 28.38 MPa and 29.15 MPa, respectively. The bending strength decreased after the addition of CaCO_3_, and the largest decrease was approximately 25%, when CaCO_3_ with a mass fraction of 50% was added.The present study revealed that the durability of MMA and MMA modified by PCE repair materials is good. After the addition of HS-770 light stabilizer with a mass fraction of 0.4%, the retention ratios for the bending strength of materials with the ratios of P0 and P3 after 1440 h of ultraviolet irradiation could reach up to 91.11% and 89.94%, respectively. The retention ratios for the bending strength of the two ratios of P0 and P3 could reach above 95% after soaking in the NaCl solution with a mass fraction of 5%, and the MgSO_4_ solution with a mass fraction of 5%, for 90 days. After the addition of the polypropylene fiber with a mass fraction of 1.5%, the retention ratios for the bending strength of the two ratios of P0 and P3 after 300 thermal shock cycles could reach up to 87.01% and 86.74%, respectively. Furthermore, the retention ratios for the bending strength of the two ratios of P0 and P3 could reach up to 97.29% and 95.13%, respectively, after 200 freeze-thaw cycles.

## Figures and Tables

**Figure 1 materials-14-00859-f001:**
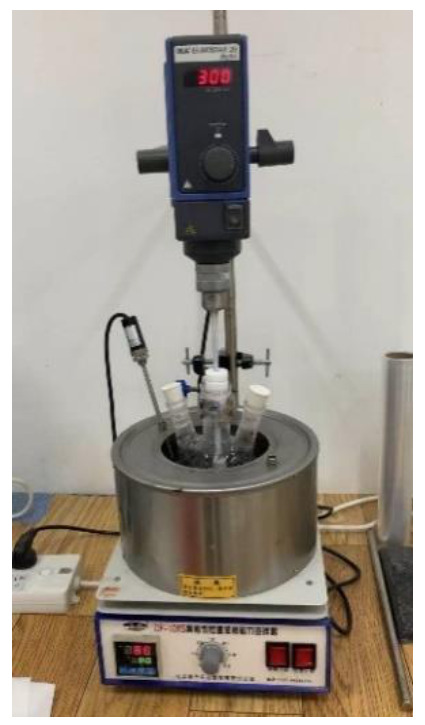
Preparation device for the MMA repair material.

**Figure 2 materials-14-00859-f002:**
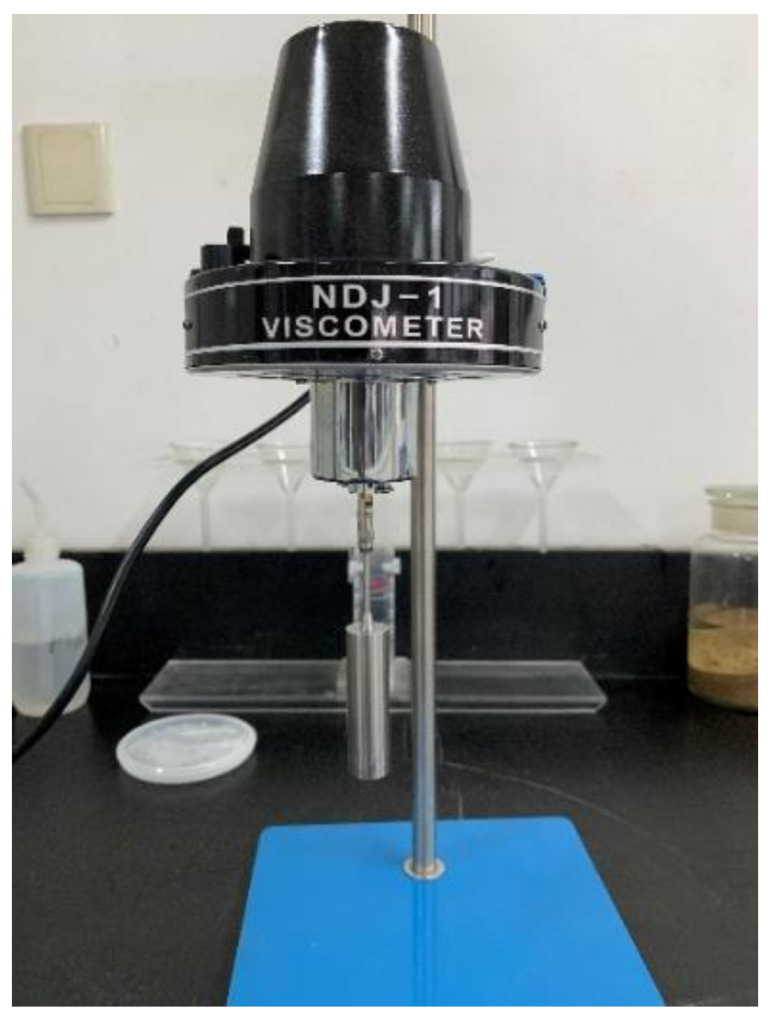
Pointer rotary viscometer NDJ-1.

**Figure 3 materials-14-00859-f003:**
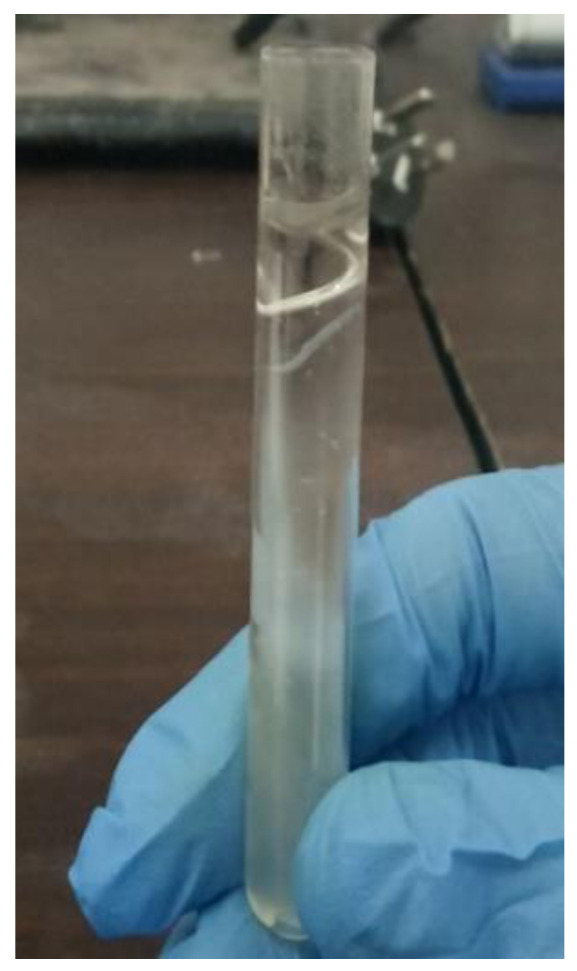
MMA repair material.

**Figure 4 materials-14-00859-f004:**
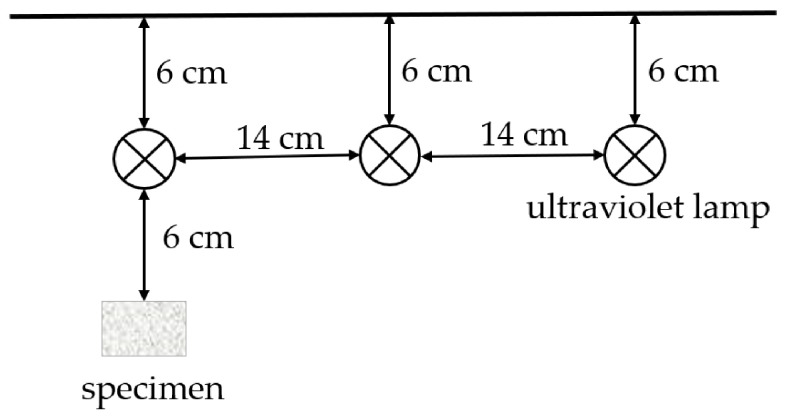
Schematic diagram of the ultraviolet resistance experiment.

**Figure 5 materials-14-00859-f005:**
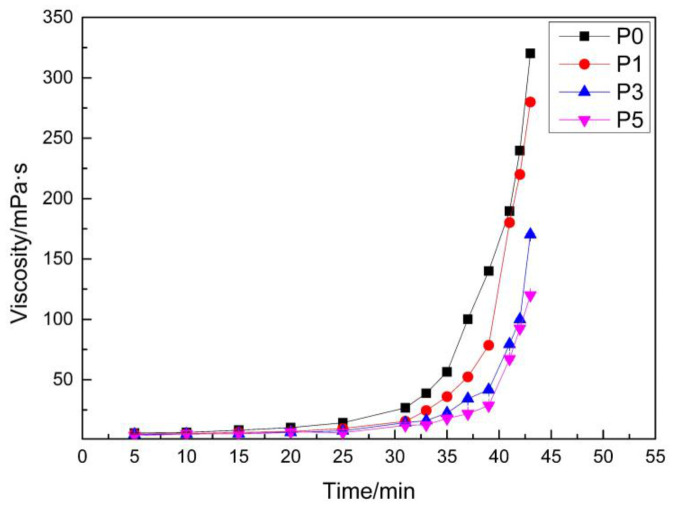
The relationship between viscosity and time of different ratio systems.

**Figure 6 materials-14-00859-f006:**
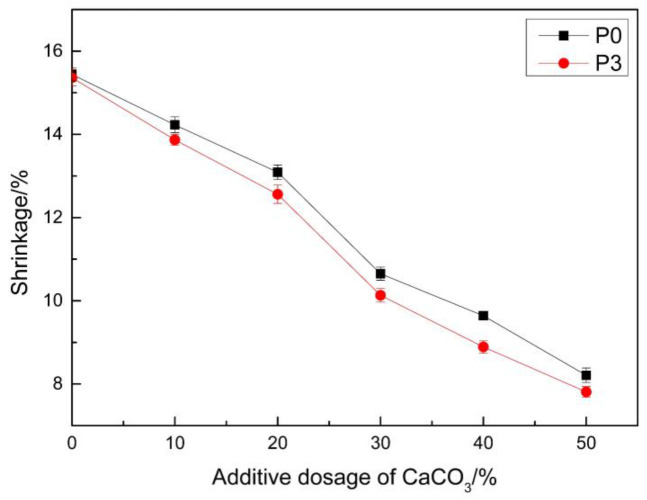
Effect of the different ratios of CaCO_3_ on shrinkage.

**Figure 7 materials-14-00859-f007:**
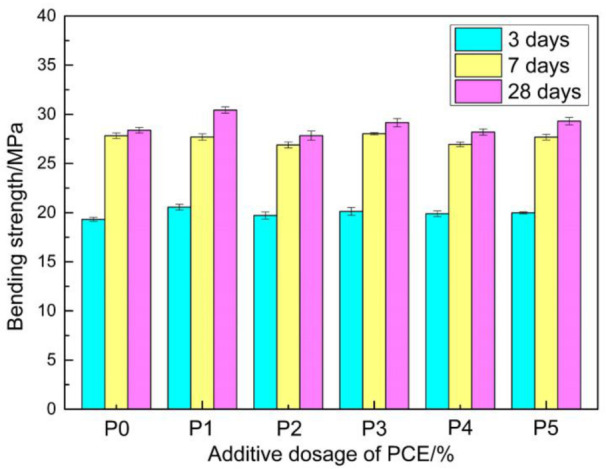
Bending strength of samples with different ratios of PCE.

**Figure 8 materials-14-00859-f008:**
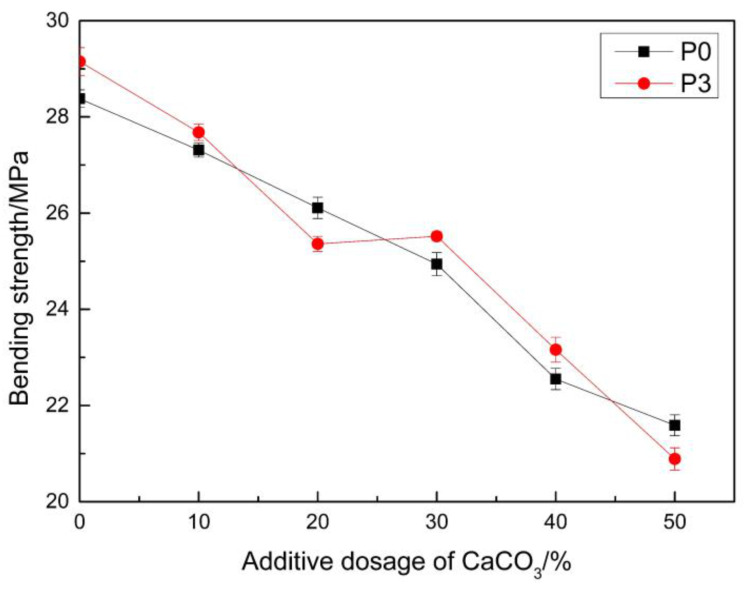
Effect of different CaCO_3_ ratios on the bending properties of materials.

**Figure 9 materials-14-00859-f009:**
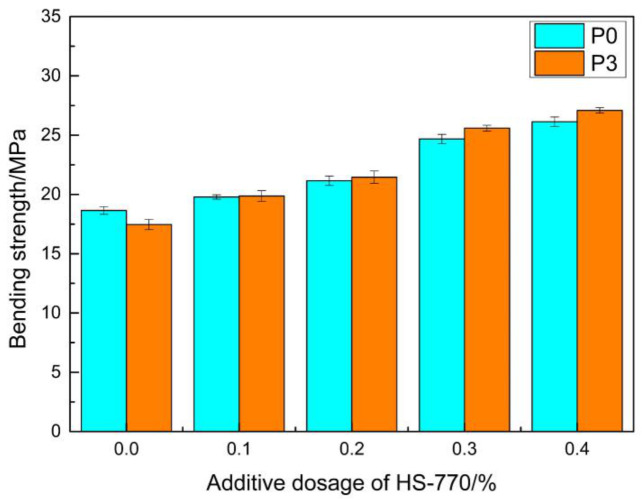
The effect of HS-770 content on bending strength.

**Figure 10 materials-14-00859-f010:**
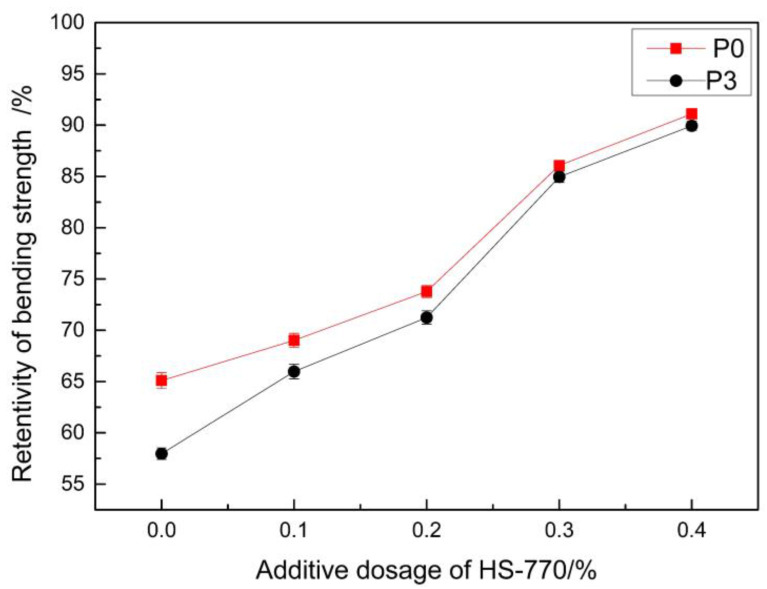
The effect of the light stabilizer on the strength retention ratios of the repair materials.

**Figure 11 materials-14-00859-f011:**
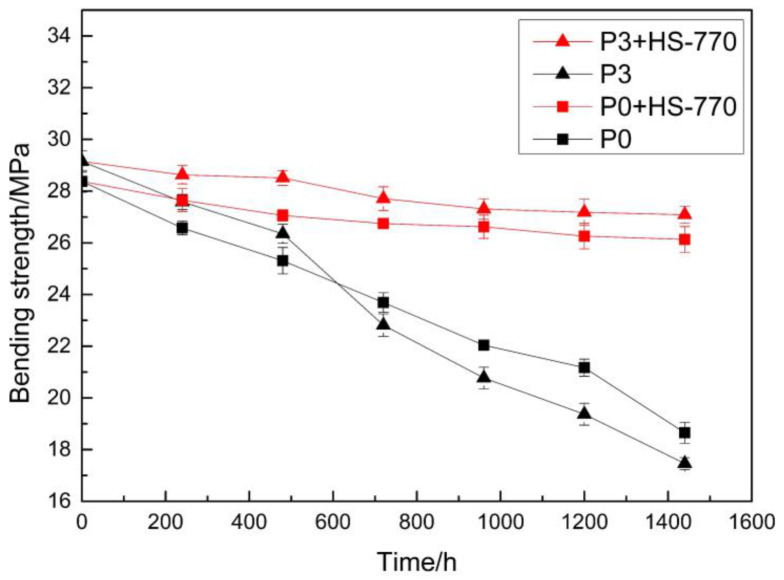
The effect of ultraviolet radiation time on the bending properties of the materials.

**Figure 12 materials-14-00859-f012:**
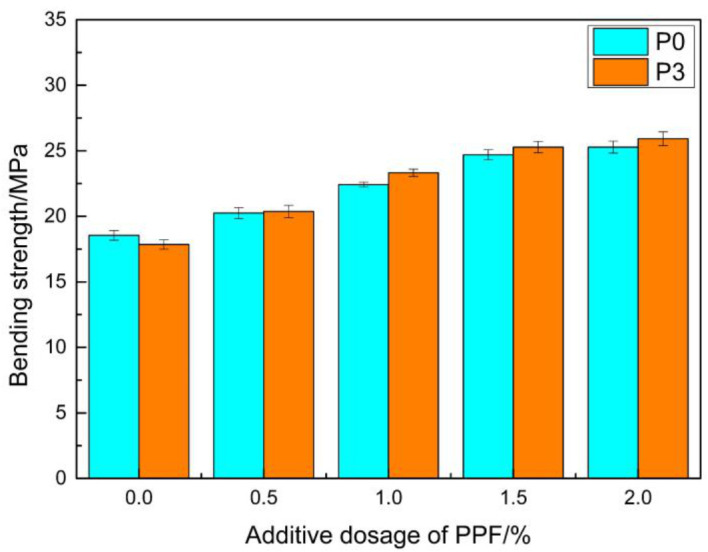
The relationship between the polypropylene fiber (PPF) content and bending strength after thermal shock.

**Figure 13 materials-14-00859-f013:**
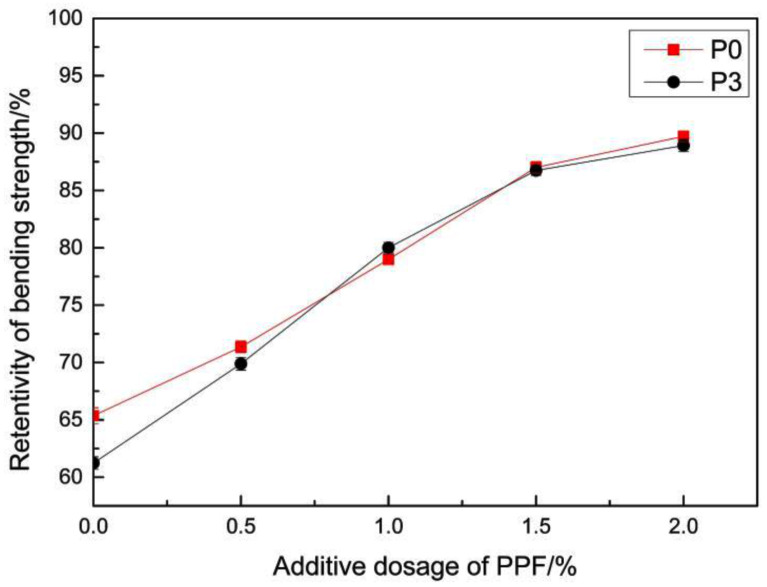
Relationship between strength retention ratios and PPF content.

**Table 1 materials-14-00859-t001:** Relevant information for the 1500 mesh heavy calcium carbonate.

Content of CaCO_3_% ≥	Content of Moisture% ≤	Content of Fe_2_O_3_% ≤	Content of SiO_2_% ≤	Average Size/μm	Specific Surface Area cm^2^/g ≤	Whiteness ≥
98.80	0.40	0.02	0.01	4.20	16,000.00	94.00

**Table 2 materials-14-00859-t002:** Raw materials ratio. MMA: Methyl methacrylate. PCE: perchloroethylene.

NO.	MMA/g	Initiator/g	Plasticizer/g	Accelerator/g	PCE/g
P0	100.00	0.60	30.00	0.60	0
P1	100.00	0.60	30.00	0.60	5.00
P2	100.00	0.60	30.00	0.60	7.50
P3	100.00	0.60	30.00	0.60	10.00
P4	100.00	0.60	30.00	0.60	12.50
P5	100.00	0.60	30.00	0.60	15.00

**Table 3 materials-14-00859-t003:** Effect of the different ratios of CaCO_3_ on viscosity.

Proportion of CaCO_3_/%	0	10	20	30	40	50
Viscosity of P0/mPa·s	90	160	290	460	580	760
Viscosity of P3/mPa·s	80	170	310	500	610	750

**Table 4 materials-14-00859-t004:** Chemical erosion resistance of the MMA repair material.

NO.	Three Days Weight Gain/%			90 Days Weight Gain/%		
	Clear Water	5%NaCl Solution	5%MgSO_4_ Solution	Clear Water	5%NaCl Solution	5%MgSO_4_ Solution
Blank sample	1.24	1.13	1.37	12.79	2.72	3.82
P0	0.47	0.34	0.41	3.03	1.34	1.72
P3	0.42	0.37	0.39	3.12	1.27	1.81

**Table 5 materials-14-00859-t005:** Chemical corrosion resistance of the repair materials.

NO.	Bending Strength Retention Ratio/%		
	Clear Water	5%NaCl Solution	5%MgSO_4_ Solution
P0	98.27	97.34	98.38
P3	96.62	96.13	99.12

**Table 6 materials-14-00859-t006:** Bending strength retention ratios for repair materials after thermal shock.

NO.	Initial Bending Strength/MPa	Bending Strength after Thermal Shock/MPa	Strength Retention Ratio/%
P0	28.38	18.55	65.36
P3	29.15	17.85	61.23

**Table 7 materials-14-00859-t007:** The freeze–thaw cycle resistance of the repair materials.

Ratio	Freeze Thaw Cycles Zero Times	Freeze Thaw Cycles 50 Times		Freeze Thaw Cycles 200 Times	
	Bending Strength/MPa	Bending Strength/MPa	Bending Strength Retention Ratio/%	Bending Strength/MPa	Bending Strength Retention Ratio/%
P0	28.38	28.04	98.8	27.61	97.29
P3	29.15	29.02	99.55	27.73	95.13

## Data Availability

All data in this article are listed in this article.The data presented in this study are available on request from the corresponding author.

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
