# Peer review of "Effect of PCE on Properties of MMA-Based Repair Material for Concrete"

_materials, 2021, doi:10.3390/ma14040859_

Round 1

Reviewer 1 Report

The manuscript is interesting and well written. No major remarks. The authors are urged to consider the following remarks in a revised submission of the manuscript:

  1. While this is not a problem, Table 1 could be omitted and replaced by text, as only PCE changes between different mixes.
  2. After the formulas please replace “—— ” with “is” or “:”
  3. Some photos of the experimental procedure (specimen preparation, experimental layout, etc.) would be needed for reference.
  4. Why are only P0, P10, P20 and P30 shown in Fig. 1?
  5. It is not clear why the specimens P0 and P20 were selected for most of the tests performed (also illustrated in Fig. 4, 6, 7, 8, 9, 11, and Tables 3, 5, 6).
  6. A thorough check for standard English and clarity of expression is needed.

Author Response

Dear Reviewer:

Thank you for the reviewer’s comments concerning our manuscript entitled “Effect of PCE on properties of MMA-based repair material for concrete”. Those comments are all valuable and very helpful for revising and improving our paper, as well as the important guiding significance to our researches. We have studied comments carefully and have made correction which we hope meet with approval. Revised portion are marked in red in the paper. The main corrections in the paper and the responds to the reviewer’s comments are as flowing:

Point 1: While this is not a problem, Table 1 could be omitted and replaced by text, as only PCE changes between different mixes.

Response 1: Thanks for the comment of reviewer. You are right. In the different proportions of the whole experiment, only the amount of PCE incorporation is different and the other amount of incorporation is the same, so Table 1 can be replaced by text. However, we have carefully considered that if the table is replaced by text, it is necessary to write the amount of PCE incorporation in each proportion and the number corresponding to the amount of incorporation should also be written into the text. This part of the content seems complicated. Our personal feeling is that it still looks simple with the table. Thank you for your valuable comment.

Point 2: After the formulas please replace “—— ” with “is” or “:”.

Response 2: Thanks for the comment of reviewer. We have replaced "--" with ":" in the manuscript as shown below.

2.3.2. Shrinkage

Here:

S: the shrinkage of the sample, %;

m1: the quality of the sample after curing, g;

V1: the volume of the sample after curing, mL;

m01: the mass of the small tube, g;

m02: the mass of the small test tube + the sample, g;

V2: the volume of the small test tube, mL.

2.3.3. Bending strength

Here:

fm: the ultimate bending strength of the material, MPa;

Fmax: the maximum load of bending failure of the material, N;

L: the distance of force between two points of the material, m;

b: the width of the section of the material specimen, m;

h: the height of the section of the material specimen, m.

Point 3: Some photos of the experimental procedure (specimen preparation, experimental layout, etc.) would be needed for reference.

Response 3: Thanks for the comment of reviewer. The pictures in the manuscript are indeed a little inadequate. We have added some equipment, experimental simulation diagram and photos of sample preparation in the manuscript.

Point 4: Why are only P0, P10, P20 and P30 shown in Fig. 1?

Response 4: Thanks for the comment made by the reviewer. We would like to make an explanation on this point. In the experiment, samples with six ratios of P0, P10, P15, P20, P25 and P30 were prepared, but only the viscosity changes of four ratios of P0, P10, P20 and P30 were shown in the viscosity test. On the one hand, the viscosity gap between adjacent ratios at the same time is not very obvious. If the viscosity of the six groups of samples is shown in the figure, it will appear that a part of the data points on the figure are very dense, so that the change trend of the whole data is not clear. On the other hand, selecting these four ratios can basically replace all the experiments to explain the content that the article intended to express, that is these four ratios are representative. To sum up, we finally selected P0, P10, P20 and P30 in the manuscript.

Point 5: It is not clear why the specimens P0 and P20 were selected for most of the tests performed (also illustrated in Fig. 4, 6, 7, 8, 9, 11, and Tables 3, 5, 6).

Response 5: Thanks for the comment made by the reviewer. It is our fault that we did not express this clearly in the manuscript, because some studies have shown that adding PCE into MMA can reduce the shrinkage. When the mass of PCE is 10 % of the mass of MMA, that is equivalent to the ratio of P20 in the present experimental study, the shrinkage rate is the lowest in all the test results. Therefore, the blank sample and the ratio of P20 are selected in the following performance test to analyze whether other properties are good or bad and whether they meet the requirements as concrete repair materials under the condition of the lowest shrinkage.

Point 6: A thorough check for standard English and clarity of expression is needed.

Response 6: Thanks for the comment made by the reviewer. We will listen carefully to your suggestions and revise the manuscript. We have had the manuscript polished with a professional assistance in writing and review it again before submission.

Special thanks to you for your good comments. We tried our best to improve the manuscript and made some changes in the manuscript. These changes will not influence the content and framework of the paper. We marked all the changes in red in revised paper. We appreciate for Reviewers’ warm work earnestly, and hope that the correction will meet with approval. Once again, thank you very much for your comments and suggestions.

Reviewer 2 Report

The overall merit of the Manuscript is very good. In the introduction most of the necessary information were included. However there is lack of some other research describing similar topic.

There is also lack of comparing results obtained by the authors with similar in the scientific field which is important in Such experiments. 

In Such article the Reviewer would like to See a scheme and a picture of the testing stand. It is always beneficial to understand how some values were measured. 

And last remark is rather the question. Why authors in the first part of the article present different samples but in the second part only refer to p20 on charts and tables. 

Overall merit as it was mentioned is very good. But manuscript need some improvements before publishing.

Author Response

Dear Reviewer:

Thank you for the reviewer’s comments concerning our manuscript entitled “Effect of PCE on properties of MMA-based repair material for concrete”. Those comments are all valuable and very helpful for revising and improving our paper, as well as the important guiding significance to our researches. We have studied comments carefully and have made correction which we hope meet with approval. Revised portion are marked in red in the paper. The main corrections in the paper and the responds to the reviewer’s comments are as flowing:

Point 1: The overall merit of the Manuscript is very good. In the introduction most of the necessary information were included. However there is lack of some other research describing similar topic.

Response 1: Thanks for the comment made by the reviewer. As you said, most of the concrete repair materials studied by scholars at home and abroad are inorganic repair materials, and epoxy resin is the main research in organic aspect. The biggest difference between methyl methacrylate in the present study and epoxy resin is that it has low viscosity and good fluidity, which can be used for the study of micro cracks. However, there are few studies on it as a concrete repair material. At present, the existing studies on MMA repair materials at home and abroad have been put forward in the introduction.

Point 2: There is also lack of comparing results obtained by the authors with similar in the scientific field which is important in Such experiments.

Response 2: Thanks for the comment of the reviewer, the manuscript did not consider the comparison with similar results in the field of science and we have now expressed the comparison results in the manuscript as shown below.

3.2. Shrinkage

Li X [31] wanted to add sodium acetate to MMA for shrinkage modification. When the mass fraction of sodium ace-tate was 3% of the MMA mass, the shrinkage rate was 18%. Compared with that, the effect of shrinkage reduction in the present experimental study was more obvious.

3.3. Bending strength

The flexural strength of the PMMA mortar prepared by Mun K J and Choi N W was within 25-35 MPa [30], which is similar to the flexural strength of the repair material studied in the present experiment.

3.4. Ultraviolet aging resistance

The strength retention rate of the epoxy mortar repair materials studied by Liu F can reach up to approximately 86% under the condition of ultraviolet irradiation for 1,000 hours [44]. These two repair materials are similar in terms of anti-ultraviolet aging performance, and the performance is relatively excellent.

3.5. Chemical corrosion resistance

The water glass suspension double-liquid grouting material studied by Wang H X was soaked in the Na2SO4 solution after 180 days. The mass loss rate was 5%, while the strength loss rate was approximately 25% [45]. It can be observed from the comparison of the data that the MMA repair material studied in the present experiment has good chemical erosion resistance.

3.6. Thermal shock aging resistance

Han Y F [14] conducted a thermal shock resistance cycle test on the modified epoxy resin, and the strength retention rate after 28 days of testing was 67.52%. By comparing these two materials, it was found that the thermal shock aging resistance of both materials was poor.

3.7. Thermal shock aging resistance

Han Y F [14] tested the frost resistance of modified epoxy resin, and the strength retention rate was 60.68 % after testing for 15 days. Compared with that, the frost resistance of the MMA repair material was extremely excellent.

Point 3: In such article the reviewer would like to see a scheme and a picture of the testing stand. It is always beneficial to understand how some values were measured.

Response 3: Thanks for the comment of the reviewer. The pictures in the manuscript are indeed a little inadequate. We have added some equipment, experimental simulation diagram and photos of sample preparation in the manuscript. The explanation of the measurement data is indeed my negligence. Now the content of the measurement value has been clearly explained in the manuscript as shown below. We believe that the addition of this part of content will further improve the quality of the article.

2.3.3. Bending strength

As a material for repairing cracks, the main mechanical property was the bending strength [35]. After the accelerator with a mass fraction of 0.5% was added, the prepolymer was evenly stirred and poured into a mold, with a size of 100×15×5 mm, and cured at 60°C. Then, the strength test was carried out on the CSS-44020 universal testing machine, and the maximum load of bending failure of the material was recorded. The bending strength of the repaired material was calculated according to Formula (2). The samples of each ratio were measured for three times, and the average value was calculated.

                           (2)

Here:

fm: the ultimate bending strength of the material, MPa;

Fmax: the maximum load of bending failure of the material, N;

L: the distance of force between two points of the material, m;

b: the width of the section of the material specimen, m;

h: the height of the section of the material specimen, m.

2.3.4. Durability

The durability of the repair material included the ultraviolet aging resistance, chemical erosion resistance, thermal shock aging resistance, and frost resistance. The first study was the ultraviolet aging resistance. Different proportions of the bis (2,2,6,6-tetramethyl-4-piperidyl) sebacate (HS-770) type light stabilizer were added into the samples of P0 and P3. Then, the samples were uniformly stirred, poured into a 100×15× 5 mm mold, and cured at 60°C. Afterwards these were placed in an ultraviolet box for 1,440 hours, taken out to test using the CSS-44020 universal testing machine, and the bending strength was calculated according to Formula (2-2). Next, these were compared with the blank specimens to obtain the retention ratios of the bending strength. Each group of samples was measured for three times and the average value was calculated. The position of the ultraviolet lamp and sample specimen in the ultraviolet box is shown in Figure 4.

Good chemical corrosion resistance can make the repair material firmly bond with the old concrete in a humid environment, and this can also prevent the water and various ions from further erosion of the repair materials [36]. In the experiment, a standard cement mortar ratio with a water-cement ratio of 0.5 and a mortar ratio of 1:3 was used to mold a size of 20×20×20 mm. Then, the P0 and P3 repair materials were applied to six surfaces of the mortar, with a thickness of approximately 1 mm. Afterwards, the samples were placed into water, the NaCl solution with a mass fraction of 5%, and the MgSO4 solution with a mass fraction of 5%. Next, these were taken out and weighed after a certain period of time to calculate the change in quality before and after the immersion. The chemical corro-sion resistance of the MMA-based repair material was evaluated by comparing this with the blank samples. Each group of samples was measured for two times, and the average value was calculated.

The samples of P0 and P3 were prepared and poured into a mold of 100×15×5 mm. Then, the thermal shock cy-cle experiment was carried out. The thermal shock cycle experiment designed for the present study was carried out according to the following steps. After heating at 105°C for 30 minutes, the repaired materials were immediately taken out and placed in a -20°C freezer for 30 minutes as a thermal shock cycle [37]. Then, the specimens were tested using the CSS-44020 universal testing machine, and the bending strength was calculated according to For-mula (2-2). Afterwards these were compared with the blank specimens to obtain the retention ratios of the bending strength after 300 cycles. Each group of samples was measured for three times, and the average value was calculated.

Concrete matrix and repair materials are often damaged due to freezing and thawing. Hence, the ability of the anti-freeze-thaw cycle should be taken into account in the preparation of the repair material [38]. The freeze-thaw cycle experiment designed for the present study was carried out according to the following conditions. The freez-ing-thawing temperature ranged from -20°C to 20°C, and the freezing-thawing time was 4-hour each time. The specimens were tested using the CSS-44020 universal testing machine, and the bending strength was calculated ac-cording to Formula (2-2), with cycles of 50 times and 200 times, in order to determine the frost resistance. Each group of samples was measured for three times and the average value was calculated.

Point 4: And last remark is rather the question. Why authors in the first part of the article present different samples but in the second part only refer to p20 on charts and tables.

Response 4: Thanks for the comment made by the reviewer. It is our fault that we did not express this clearly in the manuscript, because some studies have shown that adding PCE into MMA can reduce the shrinkage. When the mass of PCE is 10 % of the mass of MMA, that is equivalent to the ratio of P20 in this experimental study, the shrinkage rate is the lowest in all the test results. Therefore, the blank sample and the ratio of P20 are selected in the following performance test to analyze whether other properties are good or bad and whether they meet the requirements as concrete repair materials under the condition of the lowest shrinkage.

Point 5: Overall merit as it was mentioned is very good. But manuscript need some improvements before publishing.

Response 5: Thanks for the comment made by the reviewer. We will listen carefully to your suggestions and revise the manuscript. We have had the manuscript polished with a professional assistance in writing and review it again before submission.

Special thanks to you for your good comments. We tried our best to improve the manuscript and made some changes in the manuscript. These changes will not influence the content and framework of the paper. We marked all the changes in red in revised paper. We appreciate for Reviewers’ warm work earnestly, and hope that the correction will meet with approval. Once again, thank you very much for your comments and suggestions.

Reviewer 3 Report

Properties of the modified MMA-based repair material for concrete structure are tested and analyzed in the submitted manuscript. The paper is well written, organized and I believe interesting for Materials readers. The conducted experiments and tests are properly chosen for such type of research. Because of high quality of the manuscript, I recommend its publication as following minor amendments will be performed:

  1. Information on measuring uncertainty of the applied test methods is missing, it must be provided or statistical evaluation of the measured data must be done;
  2. Information of number of samples used in the particular tests is missing;
  3. it is not clear how the specimens dimensions in the bending strength test were chosen, I guess it would be better to use standard prisms for mortars’ testing, i.e. 160 mm × 40 mm × 40 mm,  please comment the size effect in the evaluation of the strength parameters;
  4. formatting of Eq. 1 and Eq. 2 must be improved, similarly quality of Figs. 2, 3 must be enhanced;
  5. it is not clear, what materials were used as initiator, plasticizer, curing agent, it must be completed and clearly described in section 2.1 Raw materials to have possibility follow the conducted experiments and prepare the same materials;
  6. information on CaCO3 is missing in section 2.1 – please complete and give at least XRF composition of this material.

Author Response

Dear Reviewer:

Thank you for the reviewer’s comments concerning our manuscript entitled “Effect of PCE on properties of MMA-based repair material for concrete”. Those comments are all valuable and very helpful for revising and improving our paper, as well as the important guiding significance to our researches. We have studied comments carefully and have made correction which we hope meet with approval. Revised portion are marked in red in the paper. The main corrections in the paper and the responds to the reviewer’s comments are as flowing:

Point 1: Information on measuring uncertainty of the applied test methods is missing, it must be provided or statistical evaluation of the measured data must be done;

Response 1: Thanks for the comment of the reviewer. As for the uncertainty, it is our fault that the uncertainty has not been taken into account. After modification, we have indicated the uncertainty of the measured data by adding error bars in the manuscript.

Point 2: Information of number of samples used in the particular tests is missing;

Response 2: Thanks for the comment of the reviewer. As for the number of samples, three samples were formed for each ratio to test the strength in the present experimental study, which has been added in the manuscript. As for the performance test, only samples with two ratios of P0 and P20 were made, and other ratios were not made. It is our fault that we did not express this clearly in the manuscript, because some studies have shown that adding PCE into MMA can reduce the shrinkage. When the mass of PCE is 10% of the mass of MMA, is equal to the ratio of P20 in this experimental study, shrinkage rate is the lowest in all the test results. Therefore, the blank sample and the ratio of P20 were selected in the following performance test to analyze whether other properties are good or bad and whether they meet the requirements of concrete repair materials when the shrinkage reaches the minimum.

Point 3: it is not clear how the specimens dimensions in the bending strength test were chosen, I guess it would be better to use standard prisms for mortars’ testing, i.e. 160 mm × 40 mm × 40 mm,  please comment the size effect in the evaluation of the strength parameters;

Response 3: Thanks for the comments put forward by the reviewer. After careful consideration and thinking, we have consulted the standards for testing the bending property of resin, and it is not suitable to use 160×40×40 mm prism, which has been modified in the manuscript to change the size of the specimen to 100×15×5 mm. The size effect is explained in the evaluation of strength parameters. Thank the reviewer for finding this problem, which also provides great help for our future research.

2.3.3. Bending strength

As a material for repairing cracks, the main mechanical property was the bending strength [35]. After the accelerator with a mass fraction of 0.5% was added, the prepolymer was evenly stirred and poured into a mold, with a size of 100×15×5 mm, and cured at 60°C. Then, the strength test was carried out on the CSS-44020 universal testing machine, and the maximum load of bending failure of the material was recorded. The bending strength of the repaired material was calculated according to Formula (2). The samples of each ratio were measured for three times, and the average value was calculated.

3.3. Bending strength

It should be noted that the size of the specimen formed in the present study was 100×15×5 mm. If the specimen size changes, it is necessary to consider the size effect. It is possible for the bending strength of the specimen to decrease with the increase in section size of the specimen.

Point 4: formatting of Eq. 1 and Eq. 2 must be improved, similarly quality of Figs. 2, 3 must be enhanced;

Response 4: Thank you for your comments. Firstly, the formula has been modified in the manuscript. Please kindly understand that there may be incompatibilities due to the different versions of Word. Secondly, as for the problems in Figure 2 and 3, it has been deleted without affecting the overall content of the article for some reasons. We apologize very much for this.

Point 5: it is not clear, what materials were used as initiator, plasticizer, curing agent, it must be completed and clearly described in section 2.1 raw materials to have possibility follow the conducted experiments and prepare the same materials;

Response 5: Thanks for the comment of the reviewer. We did not find this problem when examining the manuscript, and thank the reviewer for this problem. In this study, benzoyl peroxide (BPO) was used as an initiator, butyl phthalate (DBP) as a plasticizer and N,N-dimethylaniline (DMA) as an accelerator (not curing agent, we are sorry for the wrong expression in the manuscript), which has been supplemented in the manuscript as shown below.

2.1. Raw materials

The materials used for the present study were methyl methacrylate (C.P., Shanghai Lingfeng Chemical Reagent Co., Ltd.), benzoyl peroxide (BPO) (C.P., Shanghai Lingfeng Chemical Reagent Co., Ltd.) as an initiator, dibutyl phthalate (DBP) (C.P., Shanghai Lingfeng Chemical Reagent Co., Ltd.) as a plasticizer, N, N-dimethylaniline (DMA) (C.P., Shanghai Lingfeng Chemical Reagent Co., Ltd.) as an accelerator, and perchloroethylene (PCE) (C.P., Shang-hai Lingfeng Chemical Reagent Co., Ltd.).

Point 6: information on CaCO3 is missing in section 2.1 – please complete and give at least XRF composition of this material.

Response 6: Thanks for the comment of the reviewer. As for the CaCO3 added in the experiment, it should be made clear, thank the reviewer for finding this problem and bringing it up. But we would like to express that the CaCO3 added is a kind of heavy calcium carbonate and a product purchased, and the relevant information about this raw material has been added to the manuscript as shown below.

2.1. Raw materials

Table 1. Relevant information for the 1500 mesh heavy calcium carbonate

Content of CaCO3%≥

Content of Moisture%≤

Content of Fe2O3%≤

Content of SiO2%≤

Average size/μm

Specific surface area cm2/g≤

Whiteness

98.80

0.40

0.02

0.01

4.20

16000.00

94.00

Special thanks to you for your good comments. We tried our best to improve the manuscript and made some changes in the manuscript. These changes will not influence the content and framework of the paper. We marked all the changes in red in revised paper. We appreciate for Reviewers’ warm work earnestly, and hope that the correction will meet with approval. Once again, thank you very much for your comments and suggestions.

Round 2

Reviewer 2 Report

In Reviewer's opinion manuscript can be published.